# Dielectric Cavity-Insulator-Metal (DCIM) Metamaterial Absorber in Visible Range

**DOI:** 10.3390/nano13081401

**Published:** 2023-04-18

**Authors:** Tian-Long Guo, Fangfang Li, Matthieu Roussey

**Affiliations:** 1Center for Photonics Sciences, University of Eastern Finland, P.O. Box 111, FI-80101 Joensuu, Finland; 2Institute of Materials Technology and Engineering, Chinese Academy of Sciences, 1219 Zhongguan West Road, Ningbo 315201, China

**Keywords:** dielectric cavity, metamaterial, absorber, visible range

## Abstract

For many years, metamaterial absorbers have received much attention in a wide range of application fields. There is an increasing need to search for new design approaches that fulfill more and more complex tasks. According to the specific application requirements, design strategy can vary from structure configurations to material selections. A new combination of a dielectric cavity array, dielectric spacer, and gold reflector as a metamaterial absorber is proposed and theoretically studied in this work. The complexity of the dielectric cavities leads to a more flexible optical response than traditional metamaterial absorbers. It gives a new dimension of freedom for a real three-dimensional metamaterial absorber design.

## 1. Introduction

Metamaterial (MM) absorbers working in the visible-near infrared range have attracted much attention in recent decades. Enhanced bio/chemical sensing [1,2], energy harvesting [3], and refractive index sensing [4] is a concise list of application examples showing the great potential and value of MM absorbers. Considering the variant application aspects, for example, working frequency [5,6,7], polarization sensitivity [8], bandwidth [9,10], and thermal control [11,12], the configuration of MMs and the material selection can be different. Metal-insulator-metal MM (MIM-MM) is the most studied configuration. It achieves high absorption by establishing opposite-directional electric currents at the spacer’s top and bottom metal-dielectric interfaces. The presence of metals typically results in significant absorption of light, which generates a substantial amount of heat rather than photocarriers. Unfortunately, this can negatively impact the performance of optoelectronic devices. All-dielectric metamaterial (DMM) is another platform to achieve high absorption but without Ohmic loss. Due to the comparable strength of both electric and magnetic Mie resonance in dielectric resonators, the dielectric metasurface shows a unique property of tailoring electric and magnetic resonances individually. Huygens’ metasurface is formed when electric and magnetic resonances overlap, and the reflection from the metasurface is eliminated by destructive interference (R=0, also known as Kerker condition) [13]. Later, the high absorption (T=0) is obtained by achieving complete destructive interference between the scattered field and the incident field within each period [14,15]. Other than all-dielectric structures, configurations like dielectric metasurface on metal [16] and dielectric film on metal mirror [17] can also be applied to achieve high absorption. The combination of dielectric and plasmonic material opens a new path of electromagnetic (EM) wave modulation. However, the designs of the active element (top) of such hybrid MMs are usually limited, leading to a narrow window in terms of application. When the bottom metal reflector is determined, tuning the lateral geometry and overall structure pattern of the active element will further increase the functional versatility of the device.

All-dielectric planar multilayer is a fundamental component to manipulate EM wave propagation. When alternating the layer property rather than using periodic structure, for example, a planar microcavity, polariton splitting of intersubband transitions can be observed [18]. Moreover, when the top layer is truncated, optical surface modes (Bloch surface waves) can be found, which later can be coupled into complex systems to achieve more comprehensive applications [19]. It offers an easy wave and polarization manipulation platform with a minimum power absorption. In addition to the one-dimensional multilayer, its metasurfaces are also demonstrated to exhibit extraordinary properties. They have been studied and applied in the field of nonlinear optics, ultrafast all-optical modulation, and scattering enhancement [20,21,22].

In this work, we propose a hybrid type of MM absorber formed by substituting an all-dielectric cavity (DC) metasurface for MIM’s top active metal surface. Such a combination offers a broader range of degrees of freedom, namely the polarization, the choice of materials, and the positioning of the resonances. Based on rigorous three-dimensional (3D) numerical simulations, we detail and analyze the behavior and origin of its characteristic resonances and the spectra dependence on the incident angle under transverse electric (TE) and transverse magnetic (TM) polarized illumination. DCIM MM can sustain three excitation modes dominated by electric or magnetic resonances when excited. Moreover, a collective resonance within neighboring DC blocks can be found. Due to such unique properties from the DC array, we demonstrate that each resonance can be tuned separately. This gives more freedom for applications that require complex functions.

## 2. Computational Model

The 3D FEM model in this study was built and performed in a commercial finite element method (FEM) solver COMSOL Multiphysics with appropriate boundary conditions. The “electromagnetic waves, Frequency domain” (ewfd) interface was selected, where the electric field distribution was calculated through a full field formulation. Two cuboid perfectly matched layers (PMLs) with scattering boundary conditions on the exterior surface were placed on top and bottom of the domain to absorb any unphysical reflection at the boundaries. The overall height of the domain box was set to be 3 μm to ensure the distance between the interior boundaries of the top/bottom PML and the study object is at least one $\lambda_{\max}=800$ nm, the maximum wavelength applied in the study. Moreover, the thickness of the PML is $0.5\lambda_{\max}=400$ nm in our case. Electromagnetic waves were launched from a port boundary 20 nm under the top PML. Two additional planes (boundary monitors) were placed in the middle of the top PML-port gap and 10 nm above the bottom PML to evaluate the total reflection and transmission of the system, respectively. To establish the metamaterial in 3D, after placing the unit cell in the model center, two sets of Floquet periodic boundaries (opposite position) were placed adjacent to each other as the exterior boundaries of the physics domain in *xOy* plane. The implemented mesh is chosen in the balance of result accuracy and computation efficiency. The mesh convergence study is detailed in Appendix A and Figure A1.

The designing workflow of the unit cell is evolved from 2D cross-sectional design to 3D parameter optimization to save computational time. In particular, we first designed a 2D unit cell with high absorption under linear polarized normal illuminations in our interest range of wavelength. Typically, a TE-plane wave with **E**_*y*_ in *+y* direction, **H**_*x*_ in *+x* direction, and **k** in *−z* direction is normally illuminating on it, from the air side. Later the cuboid profile of the unit cell is selected to inherit better the optical properties that are transferred from the 2D cross-sectional design. The fine optimization of all the parameters in the 3D model is conducted by parameter sweep studies. The final implemented unit cell is shown in Figure 1a. Specifically, the top active unit is a five-layer all-dielectric cuboid cavity. The dielectric stack comprises a high refractive index material, i.e., titanium dioxide (TiO_2_), and a low index material, silicon dioxide (SiO_2_). The different layer thicknesses are $d_{{\mathrm{TiO}}_{2}}=54$ nm, $d_{{\mathrm{SiO}}_{2}}=90$ nm, and $d_{\mathrm{cavity}}=90$ nm. The lateral dimension of the cavity block (*W*) is defined by the ratio (r=0.85) of the array’s periodicity (P=430 nm). A 150-nm-thick gold layer is placed under the cavity array, and an insulating layer ($d_{{\mathrm{Al}}_{2}O_{3}}=50$ nm) is chosen as a spacer between them. The whole device is created on a glass substrate. The chosen materials are readily available in our lab, and the designed dimensions are feasible for fabrication. For clarity in the explanations, the refractive indexes of dielectric materials are set to be constant with wavelength and given for the wavelength λ=510 nm ($n_{{\mathrm{TiO}}_{2}}=2.69$, $n_{{\mathrm{SiO}}_{2}}=1.46$, $n_{{\mathrm{Al}}_{2}O_{3}}=1.77$, and $n_{\mathrm{glass}}=1.52$) [23,24,25]. The complex permittivity of gold is obtained from Rakić et al. [26] An additional small imaginary value of 7 × 10${}^{-4}$ is also applied to the refractive index of the cavity materials corresponding to the lossy nature of dielectric multilayers during fabrication [27]. The chosen parameters for the thicknesses and period are directly linked to the targeted wavelength range, i.e., the visible range. Since the resonators and their arrangement are symmetrical in *xOy* plane, the optical responses of the DCIM with normal incident TE and TM waves are identical. Appendix B and Figure A2 provide a detailed explanation of this property. For simplicity, by default, the study is performed using TE excitation unless otherwise specified.

## 3. Results

We first investigate the optical responses of the proposed absorber by solving absorption (*A*), reflectance (*R*), and transmittance (*T*) over the visible range from 500 nm to 800 nm in the aforementioned 3D FEM model. The spatially dependent, time-averaged dissipative power density is first calculated using $q_{\mathrm{abs}}=\frac{1}{2}\varepsilon_{0}\omega \varepsilon_{r}^{\prime \prime }\left(\omega \right){\left|E(r)\right|}^{2}$, where $\varepsilon_{0}$ is the vacuum permittivity, and $\varepsilon_{r}^{\prime \prime }\left(\omega \right)$ is the imaginary part of the complex material permittivity ($\varepsilon_{r}=\varepsilon_{r}^{\prime }+i\varepsilon_{r}^{\prime \prime }$) at the angular frequency ω corresponding to the electric field vector **E(r)**. The absorption is then calculated by integrating $q_{\mathrm{abs}}$ over the specific element volume and normalized by the input power $P_{\mathrm{in}}=1$ W. $A_{\mathrm{cavity}}$ and $A_{\mathrm{Au}}$ represent the absorption in the cavity array and the gold substrate, respectively. The reflected and transmitted power ($P_{R}$ and $P_{T}$) are evaluated by integrating the power flow in +z and −z directions over the boundary monitors placed at a distance of 10 nm below the top and above the bottom perfectly matched layers (PMLs), respectively. The total reflectance (*R*) and transmittance (*T*) are then calculated by normalizing $P_{R}$ and $P_{T}$ over $P_{\mathrm{in}}$. A gold layer thicker than 150 nm, greater than the skin depth of Au, is chosen to stop all input light, and zero transmission is expected throughout the studied wavelength range, as shown by the green curve in Figure 1b. Moreover, the independently calculated *A*, *R*, and *T* follow the system power conservation rule known as A+R+T=1, suggesting that the model boundaries are properly assigned. As shown in Figure 1b, three major resonances can be found at wavelengths $\lambda_{1}=560$ nm, $\lambda_{2}=643$ nm, and $\lambda_{3}=718$ nm. Most of the input EM power is dissipated in the Au layer since the imaginary permittivity of Au is larger than that of DC materials. It is also noticeable that at $\lambda_{3}$, the absorption is relatively low (around 0.88) compared to the other situations (over 0.99). At the same time, the ratio between $A_{\mathrm{cavity}}$ and $A_{\mathrm{Au}}$ is around 0.5 at $\lambda_{3}$. This is much higher than the other two resonances, which are less than 0.1. Considering the $q_{\mathrm{abs}}$ formula utilized to calculate absorption in each resonance, such a high absorption ratio in the DC array should come from much higher electric field intensity over the cavity volume than in the cavity volume the other two cases at resonances. Compared with traditional MIM absorbers with simple cuboid metal elements as the top active layer [28,29] or even when the top array is made by metal-dielectric multilayer blocks [30], the proposed DCIM absorber shows much more complex absorption modes. It is attributed to the complex modes supported in such cavities shown in the following context.

The first two rows of Figure 2 are the |**E**| and |**H**| field distributions for the three main resonances of the DCIM. It shows three modes with high localizations in the dielectric cavity region. Each of the three modes presents a distinct field pattern with some similarities. For example, fields in the spacer and at the metal/spacer interface are much weaker than in the cavity. However, the input energies cannot be fully dissipated in the cavity because of a very small $\varepsilon_{r}^{\prime \prime }$ value in dielectric materials. At $\lambda_{1}$ and $\lambda_{3}$, the electric field is mostly trapped in TiO_2_ layers (a1, c1). At $\lambda_{2}$, two major electric field confinements can be found at the top and bottom boundaries of the central TiO_2_ layer (b1), implying magnetic field confinement at the cavity center (b2). We further explore the polarization patterns in the center layer of the cavity by examing the 3D view of the electric displacement field in *y*-direction $D_{y}$ at each resonance (a3–c3). The $D_{y}$ orientation at $\lambda_{1}$ and $\lambda_{3}$ is orthogonal to that in the $\lambda_{2}$ case. On the other hand, in the *z*-axis, an opposite polarization can be found in the cavity at $\lambda_{2}$, which is another distinct difference from the other two cases and the reason for forming the magnetic field in the cavity center. The above results suggest that it yields electric or magnetic field confinement in the cavity depending on the chosen wavelength. Moreover, at $\lambda_{3}$, an apparent collective resonance can be observed and located adjacent to the cavity (c1). At such an excitation state, both the electric and magnetic fields show a higher amplitude than in other cases.

In the following, each element of DCIM is individually investigated in our 3D model to understand further the origin and mechanism of the resonance of the full structure. Identical illumination conditions are set for all samples over the wavelength from 500 nm to 800 nm, and the reflectance spectra are shown in Figure 3. The reflectance spectrum of the DCIM structure, labeled as (iv), is presented as a reference (in red). The very fundamental element of the DC array is a free-standing dielectric multilayer stack (i) invariant in *x* and *y* directions. It shows an expected Bragg mirror-like nature since the stack is made by a high refractive index layer sandwiched by two oppositely-placed Bragg mirror periods. The actual metasurface is formed by a subwavelength structuring of the stack into periodically distributed blocks. The effective refractive index of the metasurface is then different from the multilayer stack, and the optical response is also changed (ii). One can already find all resonances of the DCIM (iv) in (ii). The dip around 500 nm is the heritage from the bandgap edge in (i). When comparing with (ii) and (iv), significant changes in the reflectance amplitude and resonance position can be observed at 602 nm (●) and 670 nm (▲), respectively. However, resonances at $\lambda_{1}$ (★) and $\lambda_{3}$ (■) remain nearly unchanged. The |**E**| distributions in *xOz* plane of (ii) is given in the bottom row of Figure 3 for each resonance to correlate the optical responses between (ii) and (iv). It is clear that |**E**| patterns at ★, ▲ and ■ in (ii) are similar to DCIM resonances at $\lambda_{1}$, $\lambda_{2}$ and $\lambda_{3}$, respectively. Such similarity implies that the top cavity array is responsible for the optical response of the DCIM absorber. In the free-standing cavity array, when at resonances, only a small portion of power is dissipated in it because the extinction coefficient of the dielectric material is small. As a result, most of the power will pass through the metasurface. When coupled with the dielectric spacer and the gold mirror, the transmitted wave is partially absorbed in metal, and the rest will be reflected into the cavity with a phase shift. At maximum absorption, the reflected wave is trapped between the cavities and the metal mirror until it is fully dissipated by resistive loss or destructive interference. Concerning the resonance represented by ●, one can observe a drastic variation in the amplitude of the dip from (ii) to (iv). This variation comes from the location of the field in the external layers of the cavity, forming a completely different mode pattern in the cavity from the other cases. Thus, it leads to a decoupled system from the same spacer and the metal mirror. When looking closely at the resonance dip at ▲, a significant blue-shifting can be found after placing the spacer and metal mirror. However, in the ★ case, the resonance dip is red-shifted when placing the spacer and metal. Although the field distributions at resonances are complex and difficult to interpret, the reason for the opposite shifting direction may come from the distinct dominating field in the center cavity of ▲ (|**H**|) and ★ (|**E**|) leading to different responses to the reflected waves from the metal mirror with a phase shift. (iii) is the scenario when the dielectric spacer is removed from the DCIM. And the influence of the dielectric spacer shall be revealed by comparing it with other geometries. Reflection differences between (ii) and (iv) suggest that the DC array is responsible for building the resonance modes. The top metasurface and metal mirror are coupled by a phase shifting among transmitted and reflected waves, while the dielectric spacer is the key to adjusting the coupling. Without it, all transmitted waves from the DC array will be reflected with 180° phase change. After interference with the waves inside the cavity, new modes are built at 583 nm, 663 nm, and 712 nm, as shown in (iii). Figure 4 is the |**E**| patterns in *xOz* plane at the above three new major resonances, and it can be seen that only resonance at 712 nm shares a similar pattern as the previous 718 nm mode in (iv). When analyzing reflectance in (ii), (iii), and (iv) together, we can find that the influence of the dielectric spacer on resonance at 718 nm is quite limited. It can be attributed to the fact that the collective resonant effect among DC resonators in this mode also contributes to the total optical response and weaken the coupling effect between the DC array and the metal mirror.

From the above results, it is clear that the spacer plays a key role in coupling between the absorber (metal layer) and the resonator (dielectric stack cavity). The influence of the spacer thickness on the DCIM optical response is presented in Figure 5. The spectrum labeled ‘0’ means no spacer is in the system. For a spacer thinner than 30 nm, the resonator and the metal are far from coupling, especially for the $\lambda_{1}$ and $\lambda_{2}$ cases, which, as mentioned above, are more dependent on the coupling between metal and resonators than at $\lambda_{3}$. When using thicker spacers, one can observe a similar trend for the three resonances of interest. Resonances are red-shifted when the spacer thickness increases. The minimum in reflection is not obtained at the same, although quite close, thickness value of the spacer for each resonance. The best case is obtained for a spacer thickness of 50 nm for the $\lambda_{1}$ resonance and 60 nm for the $\lambda_{2}$ resonances. Chen demonstrated the interference theory of a metamaterial absorber in his work and suggested that both the absorption peak (reflectance dip in our case) position and intensity will shift accordingly when the spacer thickness deviates from its optimized value [31]. It agrees well with our results, and the above results show that the dielectric spacer is critical to the absorber performance since the resonator array and metal mirror in the absorber system can be coupled or decoupled by fine-tuning the spacer thickness. Still, the spacer shows a very limited effect on $\lambda_{3}$ resonance within our variation range. Another important outcome of the optimization study is that it proves the fabrication feasibility of our DCIM in practice. The absorption variation of all three modes stays within 10% while the peak shifting is also kept within the 10 nm range when the thickness of the pacer changes from 40 nm to 60 nm. Such tolerance with layer thickness variation can be easily fulfilled with our atomic layer deposition system [32].

It is also expected that the absorption performance of DCIM will be very sensitive to incident angle due to the complexity of the resonator structure. Figure 6 addresses all three modes in DCIM and their tolerance to incident angle variation from 0° to 10° under TE and TM illumination. Keeping the same trend, $\lambda_{1}$ and $\lambda_{3}$ show similar behavior under identical illumination. Reflectance variation falls in the 0.1 range when illuminated by tilted TM waves. In contrast, the reflectance of $\lambda_{2}$ is more stable under TE waves when the incident angle changes. This is due to the fact that the electric field is confined in the cavity when resonances at $\lambda_{1}$ and $\lambda_{3}$ are built, while at $\lambda_{2}$, the magnetic field dominates in the cavity. So both $\lambda_{1}$ and $\lambda_{3}$ are more “resistive” to the direction of the electric field change than to the magnetic field, while the $\lambda_{2}$ behaves the opposite way.

Besides the phase and amplitude variations observed on the resonances, polarization is the third aspect of a signal that can provide key information on a probed particle, environment, or other light signals. Choosing a different lattice (rectangular, hexagonal) for the dielectric cavities arrangement would offer even more degrees of freedom and strongly influence polarization. It has been recently shown that an array of subwavelength metallic structures can provide strong Raman enhancement at specifically tuned wavelengths by adapting the lattice constant of the structure in two orthogonal directions [33]. Operating on the same principle as our concept of two-dimensionally patterned dielectric multilayer, we open a path to similar benefits with the additional advantage of a wider choice of materials for a broader range of applications with compatibility in the bio-medical domain.

## 4. Discussion

Driven by increasing interest in MM absorbers, we propose and investigate a new approach to design MM absorbers using dielectric cavities for the top metasurface instead of a metal array. Through rigorous 3D simulations, we unfold the optical properties of the absorber. The top DC array’s geometrical parameters and material refractive index primarily control the absorber’s feature response, which is coupled with a metal mirror by an optimized dielectric spacer. The use of dielectric materials in MM absorbers complements traditional plasmonic materials, providing a wider range of refractive index contrast that covers a broad frequency range. This approach offers a more extensive selection of material candidates, which can target various application requirements. Moreover, dielectric materials are ideal for optical applications where low Ohmic loss or high-quality factors are necessary. Our exemplary case illustrates the potential benefits of combining DC resonators with metal mirrors, showcasing a new design possibility. The absorber’s complexity arises from the targeted material combination, coupled with the DC stack’s layer thickness and position sequence, enabling multi-band absorption in a more straightforward geometric form than traditional absorbers. Most design simulations can be conducted in 2D, making the process more manageable. On the other hand, the distinctive origin mechanisms of readily established multi-band absorption are preferable for applications such as sensing, which requires multiple working windows or cross-reference measurements [34].

We would like to emphasize the crucial role played by the DC top layer during its successful integration with the metal plate. This all-dielectric metasurface configuration can be further modified by substituting different dielectric material pairs with different refractive index contrast to achieve a wide range of optical functionalities. By adjusting the geometrical parameters of the dielectric resonators, the absorber’s spectral response can be tuned to target specific applications. This approach offers a straightforward way to design and fabricate metasurfaces with multiple functionalities, which can be integrated with other optical systems to realize complex light-matter interactions. The metasurface can also be embedded in transparent media to create a ‘free-standing’ function layer, further expanding its potential applications. Furthermore, the high-quality factors of dielectric resonators can facilitate a strong coupling between light and matter, enabling the observation of various optical phenomena, including Fano resonances. This approach offers a promising avenue for developing new optical devices, such as filters, sensors, and modulators that exploit the unique properties of dielectric resonators.

It is finally worth noticing that another key advantage of the proposed absorber is its compatibility with large-scale manufacturing techniques. For example, the dielectric multilayers can be fabricated using atomic layer deposition (ALD), a well-established technique with high precision and reproducibility. ALD allows for the deposition of thin films with precise control over the thickness and composition of each layer, making it ideal for the fabrication of complex multilayer structures. Furthermore, the following metasurface can be patterned using standard lithography and etching techniques, widely used in the semiconductor industry. These techniques are reliable and cost-effective and allow for the mass production of devices with high accuracy and uniformity.

The need for integrating all photonics components is pushing research to investigate combinations of materials and structures. The aim is to find solutions suitable for integrating a chip of key components that have demonstrated in recent years an innovative and interesting approach to big challenges such as single photon emission, sensing, and quantum-enabling structures. Plasmonics is often seen as the key-enabling method for such a goal, and it has shown tremendous results with the drawback of fabrication precision, which often requires expensive equipment. In this article, we have explored another way based on using a hybrid solution combining a flat metal surface supporting a dielectric cavity stack array. Such an approach is not conventional but shows a promising way to reach better integration with relaxed fabrication. As we already explained, in addition to the demonstrated wavelength-selective absorber, the concept can be applied to numerous applications without increasing the complexity of the fabrication. Such a concept bridges the dielectric structures, for example, photonic crystal, the metallic structures, i.e., plasmonics, and the hybrid ones, i.e., the metamaterials, without the limiting factor of combining the patterning of metal and dielectric simultaneously.

## Figures and Tables

**Figure 1 nanomaterials-13-01401-f001:**
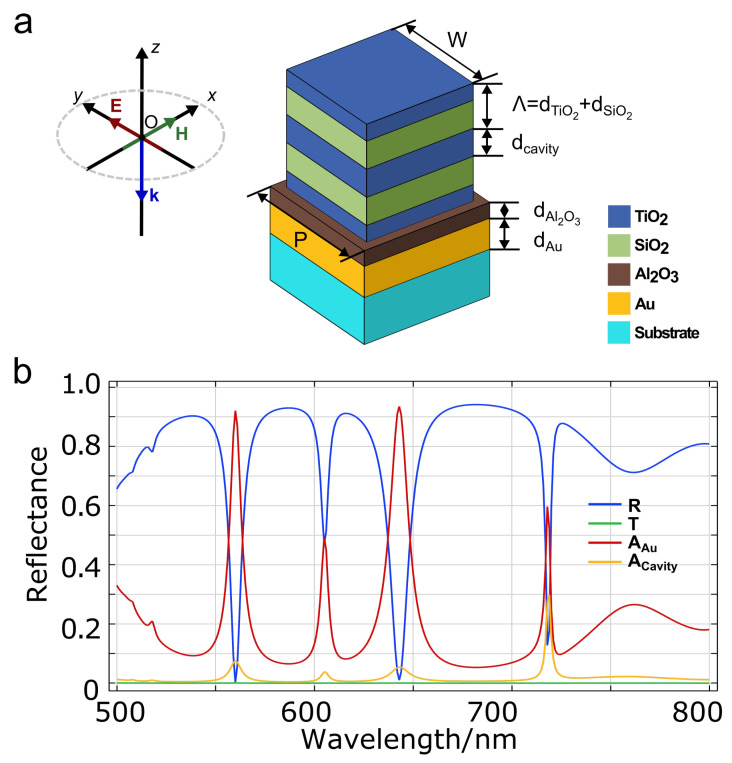
(**a**) Schematic representation of the metamaterial unit cell and a TE wave illumination in Cartesian coordinate. (**b**) Reflectance (*R*), transmittance (*T*), and absorption (*A*) spectra of the metamaterial under normal TE illumination. $A=A_{\mathrm{Au}}+A_{\mathrm{cavity}}$, where $A_{\mathrm{Au}}$ and $A_{\mathrm{cavity}}$ are evaluated in Au layer and cavity array, respectively.

**Figure 2 nanomaterials-13-01401-f002:**
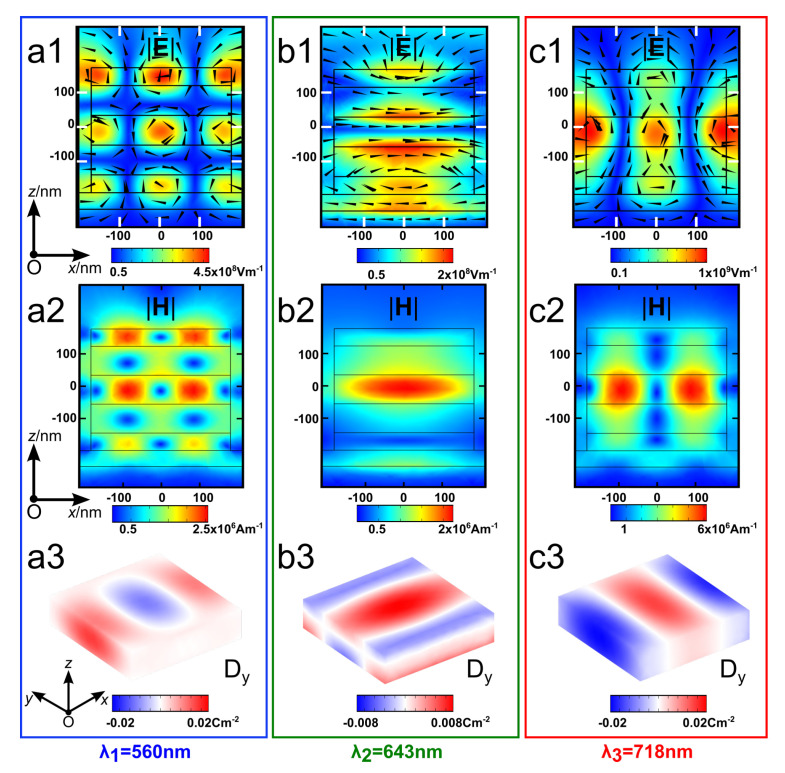
Electric field norm (|**E**|) distribution in the *xOz* plane (**a1**–**c1**), magnetic field norm (|**H**|) distribution (**a2**–**c2**), and 3D views of the electric displacement field, *y* direction ($D_{y}$) in the center layer of the cavity (**a3**–**c3**), for the three characteristic resonances $\lambda_{1}=560$ nm (first column), $\lambda_{2}=643$ nm (second column) and $\lambda_{3}=718$ nm (third column). Arrows in (**a1**–**c1**) denote corresponding magnetic field vectors.

**Figure 3 nanomaterials-13-01401-f003:**
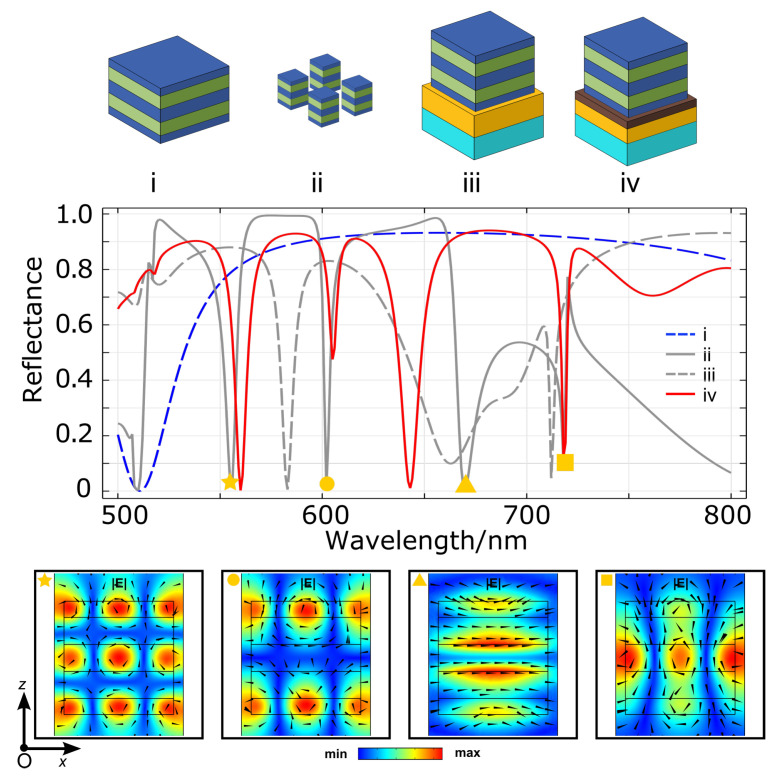
Reflectance spectra of different geometries denoted as (i) a free-standing dielectric multilayer stack with the same parameters as the dielectric cavity, (ii) a free-standing dielectric cavity array identical to DCIM top layer, (iii) DCIM MM without Al_2_O_3_ spacer, and (iv) the DCIM MM absorber. The color indicator for each material is identical to Figure 1a. 2D color maps of electric field norm (|**E**|) distribution in *xOz* plane of the sample (ii) at different resonance wavelengths labeled with geometrical markers are presented at the bottom. The arrows denote corresponding magnetic field vectors.

**Figure 4 nanomaterials-13-01401-f004:**
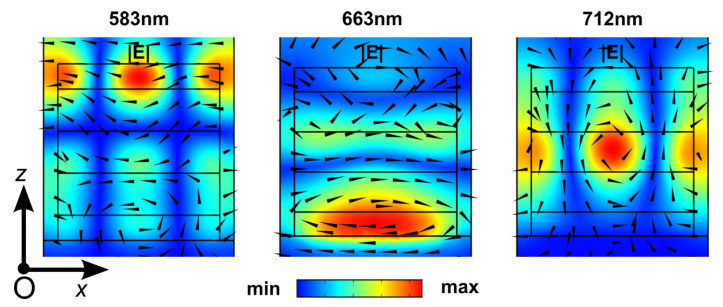
2D color maps (*xOz* plane) of electric field norm (|**E**|) distribution for resonances in metamaterial without a spacer (Figure 3 (iii)). The arrows denote corresponding magnetic field vectors.

**Figure 5 nanomaterials-13-01401-f005:**
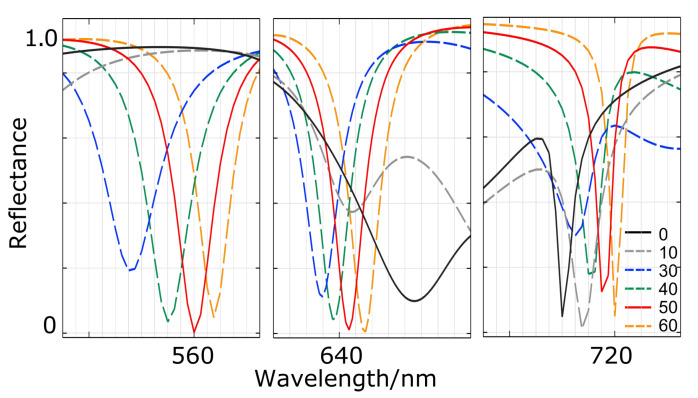
Reflectance spectra against spacer thickness variation (in nanometer) for, from left to right, resonances corresponding at $\lambda_{1}$, $\lambda_{2}$, and $\lambda_{3}$, respectively, in Figure 1.

**Figure 6 nanomaterials-13-01401-f006:**
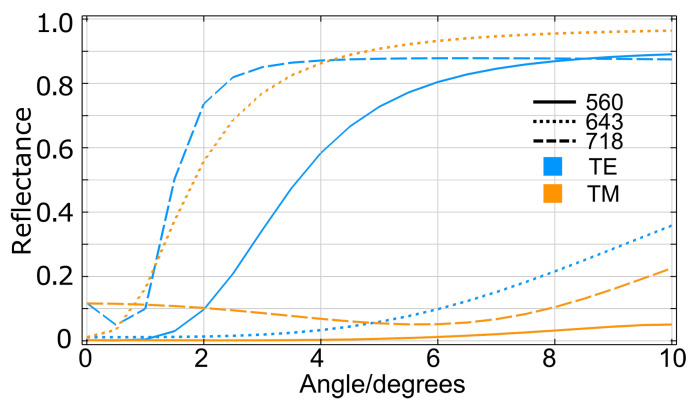
(Reflectance at $\lambda_{1}$ = 560 nm (solid lines), $\lambda_{2}$ = 643 nm (dotted lines), and $\lambda_{3}$ = 718 nm (dashed lines) with incident angle varying from $0^{\circ }$ to $10^{\circ }$. Different colors distinguish TE (blue) and TM (orange) incidence.

## Data Availability

Not applicable.

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
