# Peer review of "Dielectric Cavity-Insulator-Metal (DCIM) Metamaterial Absorber in Visible Range"

_nanomaterials, 2023, doi:10.3390/nano13081401_

Round 1

Reviewer 1 Report

This is a clearly written numerical study on DCIM absorbers. I wonder whether the introduction could be made more effective by stressing, besides the general fields of applications for absorbers, the specific choice made by the authors in the visible range, also considering that the approach selected is narrowband (is this a limitation? can the authors comment on different approaches having a better bandwidth?). Also, the application review in the introduction is somewhat repeated in the Discussion final section, this seems unnecessary. Concerning the practical implementation of the structure proposed, the authors provide in Fig. 5 a sensitivity study vs. the structure dimensions, can the authors correlate this with some specific sensitivities / accuracies of practical technological process?

Reviewer 2 Report

The paper belongs to the large array of works appeared in the last years and devoted to numerical simulations of metamaterial systems. In this case, a commercial software capable of FEM calculations is used to highlight the absorbing behavior in the visible range of a dielectric/metal system, intended as a dielectric cavity-insulator-metal architecture.

Although a possibly unexplored architecture is proposed, it is clear that such a class of papers is not expected to contain novel scientific elements, nor to provide the readers with new knowledge on a field, that of metamaterials, already well established and debated.

Contrary to most similar papers, Authors do not consider geometrical or material variants of their architecture, as a possible consequence of its complexity, opening a very large number of degrees of freedom. 

Nonetheless, I appreciated several aspects of the manuscript. It is sufficiently clear and demonstrates Authors’ efforts to find out physical (though based on numerical results) interpretations for the main findings. Moreover, even if this point should be carefully assessed in the practice, a minimal, yet sufficient, discussion is aimed at pointing out the practical feasibility of the presented system.

In summary, while I am not fully confident the work can be truly useful in view of applications, my opinion is that the paper is worth of publication and can be accepted in the present form.

Reviewer 3 Report

In this theoretical paper, the authors study capabilities of a hybrid multilayer metasurface to absorb visible light. The peculiar proposed structure consists of a dielectric resonator separated by a spacer from a gold mirror. The paper presents extensive numerical simulations of light reflection by the hybrid metasurface as a whole and by its constituent parts separately, as well as comparative analysis of the eigenstates responsible for the absorption peaks. In addition, the role of the spacer thickness is demonstrated and the absorption dependence on the angle of incidence is revealed. The paper is rather well written, the simulation methodology is clearly explained, all results look technically correct and self consistent.

Metamaterial absorbers indeed attract much attention along with other intriguing metasurface applications. In this general context, the authors’ contribution is timely and formally original. At the same time, I could not figure out from the text, which particular features are considered as positive. The authors claim that they give “a new dimension of freedom” to the absorber design, but what goals are to be achieved? It is absolutely necessary to illustrate how this can be of practical advantage. Is this rather complex structure able to outperform already known ones for a particular task?

Minor comments:

- When discussing the eigenstates responsible for the resonances in Figs. 2-4, it will be much more illustrative to show the directions of fields by arrows. Explanations in the text are lengthy and hard to follow.

- TE and TM polarizations for the normal incidence are absolutely identical for this structure. Specifying the TE polarization as in Fig. 3 caption and elsewhere is unnecessary and misleading.   

- The so-called “azimuthal rotated TE wave” sounds as nonsense. The authors mean various linear polarizations. Of course, at normal incidence, the result for them should remain identical. Fig. 6b shows this obvious fact. It can be considered as a test of the simulation precision but does not contain valuable scientific information.

Please address the first major and these three minor issues.

Round 2

Reviewer 3 Report

The authors very carefully reacted to all critical remarks. They extended the discussion of advantages of their design, improved the style of graphical presentation and shifted less important data to the Appendix. The paper is ready for publication.